# Effect of Selenium Deficiency on the Development of Overt Hepatic Encephalopathy in Patients with Chronic Liver Disease

**DOI:** 10.3390/jcm12082869

**Published:** 2023-04-14

**Authors:** Yuki Nakahata, Tatsunori Hanai, Takao Miwa, Toshihide Maeda, Kenji Imai, Atsushi Suetsugu, Koji Takai, Masahito Shimizu

**Affiliations:** 1Department of Gastroenterology/Internal Medicine, Gifu University Graduate School of Medicine, Gifu 5011194, Japan; b2111035@edu.gifu-u.ac.jp (Y.N.); miwa_t@gifu-u.ac.jp (T.M.); m0a2t1o8@gifu-u.ac.jp (T.M.); ikenji@gifu-u.ac.jp (K.I.); asue@gifu-u.ac.jp (A.S.); koz@gifu-u.ac.jp (K.T.);; 2Department of Gastroenterology, Asahi University Hospital, Gifu 5008523, Japan; 3Center for Nutrition Support & Infection Control, Gifu University Hospital, Gifu 5011194, Japan; 4Health Administration Center, Gifu University, Gifu 5011193, Japan; 5Division for Regional Cancer Control, Gifu University Graduate School of Medicine, Gifu 5011194, Japan

**Keywords:** fibrosis, hepatic encephalopathy, selenium, selenoprotein, trace element

## Abstract

Selenium is an essential trace element to maintain good health. This retrospective study investigated the prevalence of selenium deficiency and its effect on overt hepatic encephalopathy (OHE) in patients with chronic liver disease (CLD). Patients who underwent serum selenium level measurement between January 2021 and April 2022 were enrolled. The factors associated with selenium deficiency (≤10 µg/dL) and the association between selenium deficiency and OHE were analyzed. Among 98 eligible patients, 24% were observed to have selenium deficiency, with a median serum selenium level of 11.8 µg/dL. The serum selenium levels were significantly lower in patients with cirrhosis than in those with chronic hepatitis (10.9 µg/dL vs. 12.4 µg/dL; *p* = 0.03). The serum selenium levels were negatively correlated with mac-2 binding protein glycan isomer, the FIB-4 index, albumin-bilirubin (ALBI) score, and Child–Pugh score. The ALBI score remained significantly associated with selenium deficiency (odds ratio, 3.23; 95% confidence interval [CI], 1.56–6.67). During a median follow-up period of 2.9 months, nine patients experienced OHE. Selenium deficiency was associated with OHE (hazard ratio, 12.75; 95% CI, 2.54–70.22). Selenium deficiency is highly prevalent in patients with CLD and is associated with an increased risk of OHE development.

## 1. Introduction

The liver is the central organ responsible for the majority of trace element metabolism and homeostasis [1]. Selenium, an essential trace element named after Selene, the Greek goddess of the moon, plays a vital role in maintaining health through its antioxidant and anti-inflammatory properties, and aids in regulating endocrine homeostasis [1,2]. Selenium is abundant in fish, meat, and grain proteins, mainly in the form of selenium-containing amino acids, such as selenomethionine and selenocysteine. Generally, more than 90% of these amino acids are absorbed in the gastrointestinal tract and converted to selenide for the synthesis of selenoproteins in the liver [3]. As selenium is supplied to extrahepatic tissues in the form of selenoproteins, the liver plays a pivotal role in the metabolism of selenium in the human body [1,2,3].

Selenium deficiency generally occurs as a result of prolonged total parenteral and enteral nutrition without adequate selenium, hemodialysis, dilated cardiomyopathy, anorexia nervosa, and chronic liver disease (CLD) [4,5,6]. The symptoms of selenium deficiency include muscle pain and weakness, nail deformities, macrocytic anemia, and liver and thyroid dysfunctions [1]. Selenium deficiency contributes to the pathogenesis of several diseases, such as cancer, diabetes mellitus, and Keshan disease; however, it occasionally causes life-threatening events, such as irreversible cardiomyopathy, arrhythmias, and heart failure [1]. Therefore, serum selenium concentrations should be routinely measured, especially in patients at risk of its deficiency. Patients with CLD may have an increased risk of selenium deficiency; however, little is currently known about its prevalence and associated complications.

Selenium has been reported to play a key role in neurological function [6]. Therefore, selenium deficiency may contribute to the development of hepatic encephalopathy (HE), one of the most common complications of CLD that adversely affects the clinical outcomes. Selenium is critically involved in the maintenance of normal physiological functions in the form of selenoproteins, such as glutathione peroxidase, thioredoxin reductase, and selenoprotein P [7]. To date, a total of 25 selenoproteins have been identified in humans and have been implicated in anti-inflammatory activities, antioxidant and redox signaling, thyroid hormone metabolism, and immune response [1]. Since inflammation and oxidative stress are associated with the pathogenesis of HE in patients with CLD [6], it is possible to hypothesize that selenium deficiency may increase the risk of progression to overt HE (OHE).

This retrospective cohort study aimed to investigate the prevalence of selenium deficiency in patients with CLD, identify the factors associated with its deficiency, and determine the extent to which selenium deficiency is associated with an increased risk of OHE occurrence. In particular, we examined whether impaired liver function reserves, which regulate selenium homeostasis, are factors associated with selenium deficiency.

## 2. Methods

### 2.1. Study Design and Ethical Considerations

In this retrospective cohort study, 98 clinically stable patients with CLD who underwent serum selenium level measurement at Gifu University Hospital (Gifu, Japan) between January 2021 and April 2022 were retrospectively included and followed up until the first onset of OHE, last follow-up visit, or 31 July 2022, whichever occurred first.

The study was reviewed and approved by the Ethics Committee of the Gifu University Graduate School of Medicine (approval No. 2022-139), and informed consent was obtained from all participants using the opt-out approach. This study was conducted in accordance with the ethical principles outlined in the 1964 Declaration of Helsinki and its subsequent amendments.

### 2.2. Study Population

A total of 98 patients with CLD, which was defined as a progressive deterioration of liver function that lasts for more than six months, were enrolled in this study. Liver cirrhosis was diagnosed based on a combination of clinical features, histological findings, imaging features of the liver, endoscopic features of portal hypertension, and laboratory data. The degree of liver fibrosis was estimated using the serum mac-2-binding protein glycosylation isomer (M2BPGi) and FiB-4 index. The severity of liver disease was estimated using the Child–Pugh and albumin-bilirubin (ALBI) scores. Hepatocellular carcinoma (HCC) was diagnosed based on the histological features or typical imaging characteristics.

The eligible patients were 20–79 years of age, diagnosed with CLD of any etiology, and underwent serum selenium measurement. The exclusion criteria were refusal to participate, pregnant women, missing data on serum selenium measurement, a history of OHE and presence of OHE at enrollment, a history of transjugular intrahepatic portosystemic shunting, non-hepatic active malignancies, use of selenium supplements or medications, and any life-threatening comorbidities, including severe infections, and heart, respiratory, and renal failure.

### 2.3. Data Collection

The data collected at enrollment included age, sex, body mass index (BMI), etiology of CLD, presence of cirrhosis, presence of HCC, M2BPGi, FIB-4 index, ALBI score, Child–Pugh score, levels of serum albumin, creatinine, total bilirubin, zinc, ammonia, and selenium; platelet count; international normalized ratio; and branched-chain amino acid to tyrosine ratio (BTR). Blood samples were obtained after overnight fasting on the days of inpatient and outpatient visits. Serum albumin, creatinine, total bilirubin, zinc, ammonia, platelet count, and international normalized ratio were measured using clinical laboratory testing machines (JCA-BM8040; JEOL Ltd., Tokyo, Japan: cobas^®^8000; Roche Diagnostics K.K., Tokyo, Japan: XN-9100; and Sysmex Corporation, Hyogo, Japan) by standard clinical methods (Department of Clinical Laboratory, Gifu University Hospital). BTR was measured by the enzymatic method (Diacolor BTR; Toyobo, Osaka, Japan). The calculation methods of the FIB-4 index, ALBI score, and Child-Pugh score are described in Appendix A. The serum selenium levels were measured by inductively coupled plasma mass spectrometry (Agilent 7900 ICP-MS; Agilent Technologies Japan, Ltd., Tokyo, Japan). In Japan, the serum selenium measurements have been covered by insurance since 2016. In the present study, selenium deficiency was defined as serum selenium levels ≤ 10 µg/dL based on a previous report [8], and the study cohort was divided into two groups according to the selenium deficiency.

### 2.4. Outcome

The primary outcomes included the prevalence of selenium deficiency in patients with CLD, factors associated with selenium deficiency, and association between selenium deficiency and OHE development. The secondary outcomes included the characteristics of patients with and without selenium deficiency, and the association between the serum selenium levels and the degree of liver fibrosis and liver function.

### 2.5. Statistical Analysis

Continuous variables are shown as medians and interquartile ranges, and the groups were compared using the Mann–Whitney U test. Categorical variables are shown as the number of patients and percentages, and groups were compared using the chi-square test or Fisher’s exact test. The relationship between the serum selenium levels and clinical findings was analyzed using the Spearman’s rank correlation coefficient. The factors associated with selenium deficiency were determined using a multivariate logistic regression analysis with backward selection using the Akaike information criteria; the results were presented as odds ratios (ORs) and 95% confidence intervals (CIs). The predictive power of selenium deficiency for the development of OHE was assessed using the Cox proportional hazards model, and the results were presented as hazard ratios (HRs) and 95% CIs. The cumulative incidence curves for OHE were estimated using the Kaplan–Meier method, and the groups were compared using the log-rank test. All the tests were two-sided and the significance threshold was set at *p* < 0.05. Data management and analysis were conducted using JMP^®^ 14.0.0 (SAS Institute Inc., Cary, NC, USA).

## 3. Results

### 3.1. Baseline Characteristics of Enrolled Patients

The clinical characteristics of the enrolled patients are presented in Table 1. 

Among the eligible 98 patients, 56 (57%) patients were men with a median age of 73 years, 69 (70%) patients had liver cirrhosis, and 70 (71%) patients had HCC. CLD was attributed to the hepatitis B virus (14%), hepatitis C virus (6%), alcohol-related liver disease (27%), and other causes (56%). The median M2BPGi and FIB-4 index were 2.77 and 4.13, respectively. The median ALBI score was −2.36, with 37% of the patients classified as ALBI grade 1, 22% as grade 2a, 29% as grade 2b, and 12% as grade 3. The median Child–Pugh score was 5, with 72% of the patients classified as Child–Pugh class A, 19% as class B, and 8% as class C.

The median serum selenium level of all the enrolled patients was 11.8 µg/dL. The serum selenium levels were significantly lower in patients with cirrhosis than in those with chronic hepatitis (10.9 µg/dL vs. 12.4 µg/dL; *p* = 0.03).

### 3.2. Selenium Deficiency and Liver Function Reserve in Patients with CLD

In the present cohort, a total of 24% of patients with CLD were diagnosed with selenium deficiency, with median selenium levels of 8.8 µg/dL in patients with selenium deficiency and 12.6 µg/dL in those without selenium deficiency (*p* < 0.001). The patients with selenium deficiency also had a lower prevalence of HCC, higher BMI, and more advanced liver disease in terms of increased fibrosis markers, elevated ammonia levels, decreased zinc levels, and BTR than those without selenium deficiency. There were no significant intergroup differences with respect to age, sex, etiology of CLD, or renal function (Table 1).

The serum selenium levels were negatively correlated with M2BPGi (r = −0.46, *p* < 0.001; Figure 1a), FIB-4 index (r = −0.33, *p* < 0.001; Figure 1b), ALBI score (r = −0.41, *p* < 0.001; Figure 1c), and Child–Pugh score (r = −0.51, *p* < 0.001; Figure 1d). The serum selenium levels were also negatively correlated with the ammonia levels (r = −0.51, *p* < 0.001).

### 3.3. Factors Associated with Selenium Deficiency

Univariate analysis showed that factors significantly associated with selenium deficiency included BMI (OR, 1.15; 95% CI, 1.04–1.28), HCC (OR, 0,22; 95% CI, 0.08–0.58), M2BPGi (OR, 1.26; 95% CI, 1.10–1.45), ALBI score (OR, 4.05; 95% CI, 1.98–8.28), Child–Pugh score (OR, 1.58; 95% CI, 1.22–2.03), international normalized ratio (OR, 216.00; 95% CI, 11.50–4080.00), zinc (OR, 0.94; 95% CI, 0.91–0.97), ammonia (OR, 1.02; 95% CI, 1.01–1.04)), and BTR (OR, 0.62; 95% CI, 0.45–0.84). Multivariate analyses showed that only the ALBI score remained significantly associated with selenium deficiency (OR, 3.23; 95% CI, 1.56–6.67; Table 2).

### 3.4. Selenium Deficiency and OHE

During a median follow-up period of 2.9 months (interquartile range, 1.8–5.7), nine patients experienced OHE. None of the patients died or underwent liver transplantation during this period. The cumulative incidence of OHE at 3, 6, and 9 months was 4.5%, 64.2%, and 64.2% in patients with selenium deficiency, and 1.7%, 5.8%, and 10.1%, respectively, in patients without selenium deficiency. The cumulative incidence of OHE was significantly higher in patients with selenium deficiency than in those without (*p* < 0.001; Figure 2), with an HR of 12.75 (95% CI, 2.54–70.22).

In addition, we analyzed the cumulative incidence of OHE only in cirrhotic patients to clarify the relationship between selenium deficiency and the development of OHE. Among the 69 cirrhotic patients, 9 experienced OHE and the cumulative incidence of OHE was significantly higher in cirrhotic patients with selenium deficiency than in those without (*p* < 0.001; Appendix A), with an HR of 9.85 (95% CI, 2.02–48.01).

## 4. Discussion

This study is, to our knowledge, the first investigation to determine the number of patients with CLD who suffer from selenium deficiency, the clinical factors that are associated with selenium deficiency, and whether selenium deficiency can predict the occurrence of OHE. The serum selenium levels in patients with CLD have been suggested to be lower than those in healthy individuals [1]; however, few patients with CLD are screened to identify selenium deficiency because it is difficult to suspect selenium deficiency based on clinical symptoms. The findings of this study revealed that approximately a quarter of the patients with CLD had a suboptimal selenium status. The serum selenium levels decreased markedly with the progression of liver dysfunction, as assessed by the ALBI and Child–Pugh scores, irrespective of etiology. Reduced selenium levels were also closely associated with the severity of liver fibrosis, as assessed by the M2BPGi and FIB-4 index, indicating that selenium deficiency is significantly associated with impaired liver function and liver fibrosis. These results are consistent with those obtained in previous studies [4,9,10].

There are several possible explanations for the relationship between selenium deficiency and impaired liver function. One possibility is that the liver synthesizes selenoproteins that regulate selenium transport and protect hepatocytes from oxidative and inflammatory stresses [1,2,3]. Thus, impaired metabolism of selenium-containing amino acids due to hepatocyte dysfunction may reduce the serum selenium and selenoprotein levels. Another possibility is that excessive stress and viral infections associated with the pathogenesis of CLD may also interfere with selenium metabolism and synthesis of selenoproteins [11]. These possibilities may be supported by the fact that the concentration of selenium and selenoproteins increase with improved liver function through liver transplantation [9].

In addition to hepatic dysfunction, the dietary intake, urinary excretion, and intestinal absorption of selenium may be involved in the reduction of serum selenium levels in patients with CLD. The serum selenium levels are strongly correlated with selenium intake [12]; however, dietary intake in patients with CLD is reported to be inadequate to maintain optimal selenium status [13]. On the other hand, urinary selenium excretion strongly associated with selenium intake has been reported to be similar in patients with CLD as in controls, suggesting that selenium homeostasis is not maintained properly in these patients [14,15]. Selenium is absorbed primarily in the duodenum and jejunum; however, mucosal abnormalities and gut dysbiosis, which are frequently observed in patients with CLD, may impair selenium malabsorption [16]. Additionally, selenium may be deficient as a consequence of sarcopenia or malnutrition, especially in patients with CLD [15,17].

The present data support the hypothesis that selenium deficiency contributes to the occurrence of OHE in patients with CLD. Astrocytes and neurons in the brain are known to be particularly vulnerable to the oxidative and inflammatory stresses involved in the pathogenesis of HE because of their high oxygen consumption [6]. Since selenium protects cells from these stresses [6], its deficiency may increase the risk of developing OHE. Growing evidence suggests that a suboptimal selenium status is associated with a rapid cognitive decline, impaired executive functions, progressive cerebral atrophy, and encephalopathy [6,18,19,20]. Furthermore, some selenoproteins are secreted by astrocytes themselves and function as antioxidants to protect astrocytes and neurons from oxidative stress. In fact, gene expression profiling revealed that the mRNA levels of several selenoproteins were increased in the brains of patients with HE, whereas no such elevation was detected in those without HE [21].

Patients with CLD may benefit from an optimal selenium intake. Clinical and experimental data have shown that optimal selenium intake and supplementation within a safe dosage (<400 µg/day) for selenium deficiency can increase the blood selenium levels, restore antioxidant enzymes, and reduce hepatic inflammation, steatosis, and fibrosis [1]. However, a systematic review revealed that antioxidant supplements, such as selenium, may not be ineffective in treating patients with liver disease [22]. Therefore, further research is needed to establish the therapeutic effect of selenium in preventing or delaying the onset of OHE.

This study has several limitations. First, this retrospective cohort study was conducted at a single institution. In addition, the small sample size did not allow us to conclude any causality, because our findings may be affected by unmeasured confounding factors. Second, the low incidence of OHE during the observation period may not provide sufficient power to estimate the effect of selenium deficiency on the development of OHE. Third, the data regarding the selenoproteins with antioxidant and anti-inflammatory properties are lacking. Finally, the lack of data about the extent to which treatment of selenium deficiency reduces the risk of OHE in patients with CLD may lead to controversy about the therapeutic indication. There are no randomized controlled trials proving that selenium administration reduces the risk of OHE. We believe that our findings provide an important opportunity to advance the understanding of selenium in patients with CLD; however, further multicenter prospective studies taking these concerns into account, including whether selenium administration may prevent hepatic encephalopathy, are required to validate our findings.

In conclusion, the findings reported here provide compelling evidence that a suboptimal selenium status is frequently observed in patients with CLD, with more advanced liver fibrosis and disease associated with lower serum selenium concentrations. Furthermore, they shed new light on the possible relationship between selenium deficiency and OHE in patients with CLD.

## Figures and Tables

**Figure 1 jcm-12-02869-f001:**
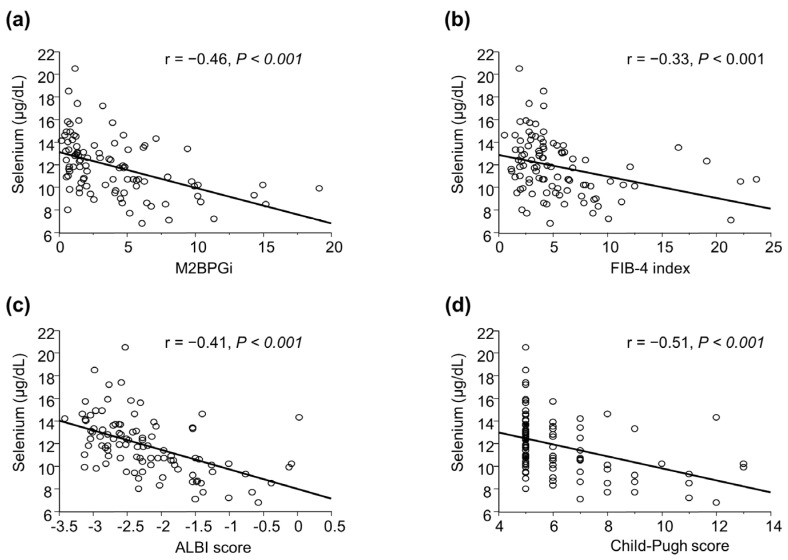
Correlation coefficients between serum selenium levels and (**a**) M2BPGi, (**b**) FIB-4 index, (**c**) ALBI score, and (**d**) Child–Pugh score in patients with chronic liver disease. The serum selenium levels were negatively correlated with M2BPGi (r = −0.46, *p* < 0.001), FIB-4 index (r = −0.33, *p* < 0.001), ALBI score (r = −0.41, *p* < 0.001), and Child-Pugh score (r = −0.51, *p* < 0.001). The data were analyzed using Spearman rank correlation coefficient. ALBI, albumin-bilirubin; M2BPGi, mac-2 binding protein glycan isomer.

**Figure 2 jcm-12-02869-f002:**
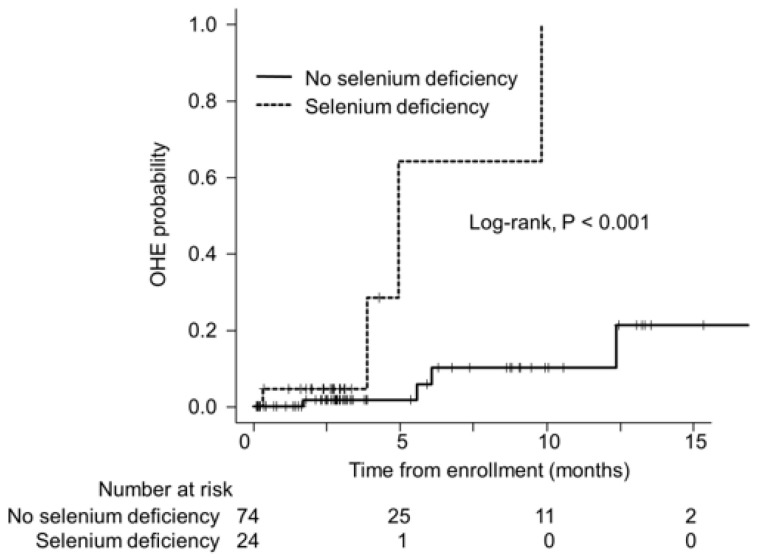
Cumulative incidence of OHE in patients with and without selenium deficiency. OHE probability was estimated using the Kaplan–Meier method and compared between the two groups using the log-rank test. OHE, overt hepatic encephalopathy.

**Table 1 jcm-12-02869-t001:** Clinical characteristics of patients with and without selenium deficiency.

Characteristics	Total Cohort (n = 98)	No selenium Deficiency (n = 74)	Selenium Deficiency † (n = 24)	*p* Value *
Age (years)	73 (66–79)	73 (66–79)	70 (63–79)	0.374
Men	56 (57)	46 (62)	10 (42)	0.098
Body mass index (kg/m^2^)	24.4 (22.2–26.9)	23.6 (21.9–25.3)	27.8 (24.0–30.1)	0.004
Etiology (HBV/HCV/Alcohol/Others)	14/6/23/55	14/4/17/39	0/2/6/16	0.081
Liver cirrhosis	69 (70)	50 (68)	19 (79)	0.279
Hepatocellular carcinoma	70 (71)	59 (80)	11 (46)	0.003
M2BPGi	2.77 (1.25–5.64)	1.83 (1.06–4.74)	5.19 (3.34–9.13)	<0.001
FIB-4 index	4.13 (2.80–6.46)	3.96 (2.74–5.93)	5.53 (3.89–8.70)	0.035
ALBI score	−2.36 (−2.78–−1.81)	−2.52 (−2.82–−2.14)	−1.52 (−2.29–−1.18)	<0.001
ALBI grade (1/2a/2b/3)	36/22/28/12	34/17/19/4	2/5/9/8	<0.001
Child–Pugh score	5 (5–7)	5 (5–6)	7 (6–9)	<0.001
Child–Pugh class (A/B/C)	71/19/8	61/10/3	10/9/5	<0.001
Albumin (g/dL)	3.7 (3.2–4.2)	3.9 (3.4–4.3)	2.9 (2.6–3.6)	<0.001
Creatinine (mg/dL)	0.75 (0.64–0.91)	0.74 (0.64–0.85)	0.82 (0.66–1.12)	0.100
Total bilirubin (mg/dL)	1.0 (0.7–1.5)	0.9 (0.7–1.3)	1.5 (1.0–1.9)	0.006
Platelet (10⁹/L)	139 (91–166)	145 (107–173)	102 (77–148)	0.051
International normalized ratio	1.00 (0.93–1.11)	0.98 (0.92–1.05)	1.15 (1.02–1.31)	<0.001
Zinc (μg/dL)	67 (57–76)	70 (61–78)	55 (42–69)	0.001
Ammonia (µg/dL)	62 (48–90)	60 (44–79)	90 (57–131)	0.002
BTR	4.8 (3.3–6.3)	5.0 (4.0–6.3)	2.9 (2.6–5.0)	0.001
Selenium (µg/dL)	11.8 (10.1–13.4)	12.6 (10.9–13.9)	8.8 (8.2–9.5)	<0.001

The values are shown as numbers (percentages) or medians (interquartile ranges). † Defined as serum selenium level ≤ 10 μg/dL. * The clinical characteristics of the two groups were compared using the chi-square test for categorical variables and Mann–Whitney U test for continuous variables. ALBI, albumin-bilirubin; BTR, branched-chain amino acid-to-tyrosine ratio; HBV, hepatitis B virus; HCV, hepatitis C virus; M2BPGi, mac-2 binding protein glycan isomer.

**Table 2 jcm-12-02869-t002:** Predictors associated with selenium deficiency in patients with chronic liver disease.

Characteristics	OR (95% CI)	*p* Value *
Body mass index, kg/m^2^	1.12 (0.99–1.25)	0.065
ALBI score	3.23 (1.56–6.67)	0.002

* Multivariate logistic regression analysis with backward selection using the Akaike information criteria. ALBI, albumin-bilirubin; CI, confidence interval; OR, odds ratio.

## Data Availability

Data sharing not applicable.

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
