# Peer review of "Effect of Selenium Deficiency on the Development of Overt Hepatic Encephalopathy in Patients with Chronic Liver Disease"

_jcm, 2023, doi:10.3390/jcm12082869_

Round 1
Reviewer 1 Report
Dear Authors,
I have received an article entitled "Effect of selenium deficiency on the development of overt hepatic encephalopathy in patients with chronic liver disease". This is an interesting article and some changes may improve this manuscript:
1) Please specify explicitly what is the study design of this manuscript such as "cross-sectional" or "retrospective cohort"
2) The authors should explain in the introduction or methods what constitutes "Factors that affect selenium deficiency". I also do not see any factors being mentioned in the results section.
3) Please explain how "M2BPGi, FIB-4 index, ALBI score, Child–Pugh score, levels of serum albumin, creatinine, total bilirubin, zinc, ammonia, and sele-97 nium; platelet count; international normalized ratio; and branched-chain amino acid to 98 tyrosine ratio (BTR)" are measured. If they were measured using machines, please explain in a greater specification of machines used and how were the samples obtained. If scoring systems were used, please attach in a supplementary material how the scores were obtained.
4) How as CLD defined in this study?
5) This paper has a 46% similarity report in Turnitin, with >10% coming from two articles that the authors previously published. Please rephrase some sentences such that they are not too similar.
a) https://www.mdpi.com/2077-0383/10/15/3448/htm
b) https://link.springer.com/article/10.1007/s00535-022-01925-0?
Author Response
Responses to Reviewer 1
Thank you very much for reviewing our manuscript and offering valuable advice. We appreciate your comments, which have helped us to improve our manuscript. Please find below detailed responses to the reviewer’s comments.
Comments
1) Please specify explicitly what is the study design of this manuscript such as "cross-sectional" or "retrospective cohort"
Thank you for your comment. This is a retrospective cohort study and we have added this description in the revised manuscript (Page 2, line 64; and Page 2, line 71).
2) The authors should explain in the introduction or methods what constitutes "Factors that affect selenium deficiency". I also do not see any factors being mentioned in the results section.
We sincerely apologize to the reviewer for the insufficient description. In the present study, we examined whether impaired liver function reserve, which regulate selenium homeostasis, are factors associated with selenium deficiency (Page 2, lines 67–68). We also described factors associated with selenium deficiency in the Results section (Page 5, lines 191–199). Thank you very much for your valuable comment.
3) Please explain how “M2BPGi, FIB-4 index, ALBI score, Child–Pugh score, levels of serum albumin, creatinine, total bilirubin, zinc, ammonia, and selenium; platelet count; international normalized ratio; and branched-chain amino acid to tyrosine ratio (BTR)” are measured. If they were measured using machines, please explain in a greater specification of machines used and how were the samples obtained. If scoring systems were used, please attach in a supplementary material how the scores were obtained.
Based on the reviewer’s comment, we have added more information about the clinical laboratory testing machines used to measure blood samples and how the samples were obtained in the Methods section. In addition, we have added more detailed information concerning scoring systems such as the FIB-4 index, ALBI score, and Child-Pugh score in the supplementary material (Page 3, lines 104–112; and new Supplementary Table 1). We appreciate your review and positive suggestions for our manuscript.
4) How as CLD defined in this study?
Thank you for your comment. CLD was defined as a progressive deterioration of liver function that lasts for more than six months. We have added this description in the revised manuscript (Page 2, lines 82–83).
5) This paper has a 46% similarity report in Turnitin, with >10% coming from two articles that the authors previously published. Please rephrase some sentences such that they are not too similar.
- a) https://www.mdpi.com/2077-0383/10/15/3448/htm
- b) https://link.springer.com/article/10.1007/s00535-022-01925-0?
Based on the reviewer’s suggestion, we have revised the manuscript accordingly. We have also requested proofreading to improve the quality of the English text (Please see attached Editing Certificate). We greatly appreciate again the reviewer’s important comment.
In closing, we would like to thank you again for your comment on improving the quality of the paper. We hope that the above responses meet with the approval of the editors and reviewers.

Reviewer 2 Report
In this manuscript, the authors investigated the prevalence of selenium deficiency and its effect on overt hepatic encephalopathy (OHE) in patients with chronic liver disease (CLD). 98 CLD patients were enrolled. The authors found that 24% of CLD patients had selenium deficiency, that the serum selenium levels were significantly lower in cirrhotic patients cirrhosis as compared with chronic hepatitis patients, that the serum selenium levels were negatively correlated with mac-2 binding protein glycan isomer, the FIB-4 index, albumin-bilirubin (ALBI) score, and Child–Pugh score, that the ALBI score was significantly associated with selenium deficiency by multivariate analysis, and that selenium deficiency was associated with OHE. So the authors concluded that selenium deficiency is highly prevalent in CLD patients and is associated with an increased risk of OHE development.
The is a retrospective study to investigate the prevalence of selenium deficiency and its effect on overt hepatic encephalopathy in CLD patients. The data collection and analysis are appropriate. The authors obtained reasonable results, and the manuscript was well prepared. Although the originality of this study was not high enough, the manuscript could provide useful information for clinicians to manage patients with chronic liver disease.
Comments
1. The Title of this manuscript is "Effect of selenium deficiency on the development of overt hepatic encephalopathy in patients with chronic liver disease ". However, the results only showed the association of selenium deficiency on the development of overt hepatic encephalopathy. To clarify the hypothesis of this manuscript, the authors should analyze the cumulative incidence of OHE only in cirrhotic patients.
2. The authors should provide the data that administration of selenium could prevent the overt hepatic encephalopathy in patients with chronic liver disease to prove the causal relationship between selenium deficiency and the development of overt hepatic encephalopathy.
3. The English needs polishing.
Author Response
Responses to Reviewer 2
Thank you very much for reviewing our manuscript and offering valuable advice. We appreciate your comments, which have helped us to improve our manuscript. Please find below detailed responses to the reviewer’s comments.
Comments
- The Title of this manuscript is "Effect of selenium deficiency on the development of overt hepatic encephalopathy in patients with chronic liver disease ". However, the results only showed the association of selenium deficiency on the development of overt hepatic encephalopathy. To clarify the hypothesis of this manuscript, the authors should analyze the cumulative incidence of OHE only in cirrhotic patients.
Thank you for your valuable comments. We analyzed the cumulative incidence of OHE only in cirrhotic patients to clarify the relationship between selenium deficiency and the development of OHE. Among the 69 cirrhotic patients, 9 experienced OHE and the cumulative incidence of OHE was significantly higher in cirrhotic patients with selenium deficiency than in those without (P<0.001; Supplementary Figure 1), with an HR of 9.85 (95% CI, 2.02–48.01). We have added this information to the revised manuscript (Page 6, lines 213–217; Supplementary Figure 1). We appreciate your suggestion that improved the quality of the manuscript.
- The authors should provide the data that administration of selenium could prevent the overt hepatic encephalopathy in patients with chronic liver disease to prove the causal relationship between selenium deficiency and the development of overt hepatic encephalopathy.
Thank you for the valuable suggestions. We agree with the reviewer’s point about the extent to which administration of selenium could prevent the development of OHE in patients with CLD. However, no randomized controlled trials exist to demonstrate that treatment of selenium deficiency reduces the risk of OHE. We agree that this is an important area that requires further research, but at present it is beyond the scope of the present study. Based on the reviewers' comments and these considerations, we have revised text (Page 8, lines 285–293). We hope to address this issue in future research. Thank you for your insightful comments.
- The English needs polishing.
Regarding the comments on the English in our manuscript, we acknowledge the importance of improving its quality. We have had the manuscript edited/proofread by a native speaker of English from a professional English editing service company. We have attached a certificate from the editing company stating that the manuscript has been professionally edited (see supplementary file). Thank you for your comment.
In closing, we would like to thank you again for your comment on improving the quality of the paper. We hope that the above responses meet with the approval of the editors and reviewers.
